# A Review of the Current Status of G6PD Deficiency Testing to Guide Radical Cure Treatment for Vivax Malaria

**DOI:** 10.3390/pathogens12050650

**Published:** 2023-04-27

**Authors:** Arkasha Sadhewa, Sarah Cassidy-Seyoum, Sanjaya Acharya, Angela Devine, Ric N. Price, Muthoni Mwaura, Kamala Thriemer, Benedikt Ley

**Affiliations:** 1Global and Tropical Health Division, Menzies School of Health Research and Charles Darwin University, Darwin 0810, Australia; 2Centre for Epidemiology and Biostatistics, Melbourne School of Population and Global Health, University of Melbourne, Melbourne 3010, Australia; 3Centre for Health Policy, Melbourne School of Population and Global Health, University of Melbourne, Melbourne 3010, Australia; 4Mahidol-Oxford Tropical Medicine Research Unit (MORU), Faculty of Tropical Medicine, Mahidol University, Bangkok 10400, Thailand; 5Centre for Tropical Medicine and Global Health, Nuffield Department of Clinical Medicine, University of Oxford, Oxford OX1 2JD, UK

**Keywords:** *P. vivax*, G6PD testing, radical cure, malaria elimination, policy, treatment guidelines, implementation, point of care diagnostics

## Abstract

*Plasmodium vivax* malaria continues to cause a significant burden of disease in the Asia-Pacific, the Horn of Africa, and the Americas. In addition to schizontocidal treatment, the 8-aminoquinoline drugs are crucial for the complete removal of the parasite from the human host (radical cure). While well tolerated in most recipients, 8-aminoquinolines can cause severe haemolysis in glucose-6-phosphate dehydrogenase (G6PD) deficient patients. G6PD deficiency is one of the most common enzymopathies worldwide; therefore, the WHO recommends routine testing to guide 8-aminoquinoline based treatment for vivax malaria whenever possible. In practice, this is not yet implemented in most malaria endemic countries. This review provides an update of the characteristics of the most used G6PD diagnostics. We describe the current state of policy and implementation of routine point-of-care G6PD testing in malaria endemic countries and highlight key knowledge gaps that hinder broader implementation. Identified challenges include optimal training of health facility staff on point-of-care diagnostics, quality control of novel G6PD diagnostics, and culturally appropriate information and communication with affected communities around G6PD deficiency and implications for treatment.

## 1. Introduction

Approximately 2.5 billion people are at risk of a *Plasmodium vivax* (*P. vivax*) infection. Between 5 and 14 million cases are recorded annually, the majority of which are reported in Southeast Asia [1,2,3]. Increased efforts of national malaria control programs (NMCPs) have led to a reduction of the global *P. vivax* burden. According to the World Health Organisation (WHO), cases decreased from 24.5 million in 2000 to 4.9 million in 2021 [3]; while Battle et al. estimated a reduction of 41.6% between 2000 and 2017, from 24.5 to 14.3 million cases [2].

Despite these advances, the control and elimination of vivax malaria is challenging, largely due to the parasite’s biology and ability to survive in the human host. *P. vivax* gametocytes develop before the onset of symptoms, resulting in asymptomatic but infectious patients who cannot be identified by passive surveillance [4,5,6]. Secondly, the sensitivity of most point-of-care (PoC) tests for *P. vivax* is lower than for *P. falciparum* due to lower parasite densities [7,8,9]. Thirdly, *P. vivax* forms dormant liver stages (hypnozoites) that can reactivate weeks to months after a primary infection [10], resulting in recurrent episodes of malaria (relapse) [2,11]. In some areas, up to 80% of clinical cases are due to relapse [12], causing long-term morbidity and significant health and economic burdens [13].

The 8-aminoquinoline (8-AQ) class of antimalarial compounds are the only available drugs with good efficacy against hypnozoites. The treatment of vivax malaria requires killing both the peripheral blood stage parasites with a blood schizontocidal and the dormant liver stages with an 8-AQ drug—together, this is referred to as radical cure. Primaquine (PQ) is the most widely used 8-AQ [14], and is usually administered over 14 days as a total dose of 3.5 mg/kg when used for radical cure in line with WHO recommendations [15]. More recently, the WHO Treatment Guidelines were updated to include a 7-day treatment regimen with the same total dosage [16]. The anti-relapse efficacy of primaquine is related to the total dose administered, and higher doses are sometimes recommended in patients at high risk of relapse [15,17]. The long treatment duration is often associated with poor adherence and low effectiveness [18,19]. Novel, short-course, PQ-based regimens with higher daily doses [20] and the recently introduced single-dose 8-AQ tafenoquine (TQ) [21] both have the potential to improve adherence. 

Whilst well tolerated in most recipients, 8-AQs can cause severe haemolysis in individuals with the common enzymopathy glucose-6-phosphate dehydrogenase (G6PD) deficiency, necessitating G6PD testing to reduce this risk and guide radical cure. The degree of drug-induced haemolysis is dose-dependent, and since shorter PQ regimens require a higher daily dose to achieve the same total dose, these regimens are associated with a higher risk of drug-induced haemolysis. PQ is rapidly eliminated (half-life 4 to 9 h), whereas TQ is more slowly eliminated (half-life ~14 days), allowing it to be administered as a single dose [22]. Due to TQ’s long half-life, patients are continuously exposed to the oxidative effects of the drug, underlining the need for reliable G6PD testing to guide treatment.

## 2. G6PD Deficiency

G6PD is a ubiquitous enzyme [23] and the rate-limiting component of the pentose phosphate pathway (PPP). In red blood cells (RBCs), G6PD is essential to maintain the cells’ redox potential by producing reduced nicotinamide adenine dinucleotide phosphate (NADPH) [24]; NADPH is a key electron donor for the conversion of oxidised glutathione (GSSG) into reduced glutathione (GSH). GSH, in turn, captures free radicals that could cause oxidative damage. Human RBCs contain neither a nucleus nor mitochondria and cannot replenish G6PD levels. Reticulocytes and young RBCs therefore have up to a 10-fold higher G6PD activity compared to older RBCs [25].

G6PD deficiency is among the most common enzymopathies affecting 400 to 500 million people [26], and it is most prevalent in current and historically malaria endemic areas [27,28,29], suggesting some form of protective effect of G6PD deficiency against malaria or a *Plasmodium* species infection [30,31]. The G6PD gene is located on the X-chromosome (Xq28), so males are either hemizygous deficient or normal. Females have two X-chromosomes, one of which is randomly deactivated at the cellular level at an early embryonic stage through a process called lyonization [32]. Accordingly, females can be homozygous deficient or normal, or heterozygous for the G6PD gene. Heterozygous females have two distinct RBC populations, a G6PD normal and a G6PD deficient one, with proportions differing depending on the degree of lyonization (Figure 1 [33]). In hemizygous, homozygous, and heterozygous deficient individuals, the risk for drug-induced haemolysis depends on the underlying G6PD genetic variant, the degree of lyonization in heterozygous females, the degree of oxidant exposure, and the age of the RBC population [23].

Phenotypic G6PD activity is measured in units per gram haemoglobin (U/g Hb). The gold standard is spectrophotometry; however, measures can differ significantly between assays, populations, and locations, confounding direct comparison [34]. Instead, activities are normalized and expressed in the percentage of normal activity, defined by the site-specific adjusted male median (AMM) calculated from quantitatively measured G6PD activity of the male population [35]. Most hemi- and homozygous individuals have G6PD activities below 30% (G6PD deficient) or above 70% to 80% of normal activity (G6PD normal) [36]. Conversely heterozygous females have enzyme activities ranging from close to 0% to almost normal activities, with activities of the majority of heterozygous females clustering around the 50% mark [37].

## 3. Overview of G6PD Test Formats and Products

There are multiple methods to diagnose G6PD deficiency [38], including molecular tests for population screening and genotyping [39], flow cytometry based methods to quantify degree of lyonization in heterozygous females [40,41], and phenotypic tests (both qualitative and quantitative) used in clinical settings for case management and reference testing.

Molecular methods identify polymorphisms of the G6PD gene that have been associated with enzyme activity [42]. Sequencing methods are able to identify known and novel G6PD variants, while variant-specific genotyping methods such as the PCR-RFLP, PCR-SSCP, or hybridisation arrays are only suitable for areas where the most common variants are known. Current molecular methods require good laboratory infrastructure and well-trained staff, and the interpretation of the final result requires specialised tools [43]. No existing molecular method can provide results within a timeframe that would render the method suitable for testing to support radical cure treatment decisions at the bedside [44]. Furthermore, even between individuals carrying the same G6PD genetic variants, G6PD activity varies [36], and it may be further confounded by changes in G6PD activity associated with acute malaria [45,46,47]. These molecular methods are employed for the surveillance of populations and research purposes only.

Cytochemical assays do not measure G6PD residual activity but can distinguish individual G6PD deficient RBCs from G6PD normal RBCs to determine the proportion of G6PD deficient cells in heterozygous females [40,41]. The method requires a costly flow cytometer, with protocols involving hazardous chemicals [40].

Phenotypic methods measure G6PD activity from blood samples by measuring NADPH production, either directly or indirectly, and either qualitatively or quantitatively. Most phenotypic assays require a laboratory (Table 1) and specialized training or experience in interpreting the results.

Current PoC phenotypic assays require minimal resources, low expertise in handling, and a short time to show a result. In recent years, several G6PD PoC tests have been developed, including lateral flow qualitative tests to identify G6PD deficiency at around 30% of normal G6PD activity as well as quantitative handheld devices (biosensors) that can identify individuals with intermediate activity (Table 1).

To guide future development of G6PD diagnostics, the WHO has created two target product profiles (TPPs) [74]; one is for screening G6PD at point of care to guide individual treatment decisions (TPP #1), and the other is a one-time quantitative assay that is meant to classify individuals as deficient, intermediate, or G6PD normal once in order to guide any future treatment decisions (TPP #2). The main acceptable and desirable characteristics of TPP #1 and TPP #2 are summarized in Table 2.

## 4. Policy and Practice of G6PD Testing 

G6PD testing is recommended as part of a ‘good practice statement’ by the WHO [75]. In practice, testing has not been implemented in endemic countries, leading to a disconnect between international guidelines and national policy and practice [76]. Barriers to implementation include the perceived low risk of drug-induced haemolysis, low prioritisation of radical cure treatment because of the perception that vivax malaria is benign, additional costs and financing of implementation, lack of clear guidance on how to provide treatment to G6PD deficient patients, and concerns over additional workload for health staff [77,78,79]. This has changed in recent years, with an increasing number of countries implementing PoC testing.

A review of current policy on G6PD testing and treatment (Appendix A) is presented according to region: Asia-Pacific (Appendix A [80,81,82,83,84,85,86,87,88,89,90,91,92,93,94,95,96,97,98,99,100,101,102,103,104,105,106,107,108,109,110]), Africa and the Middle East (Appendix A [111,112,113,114]), and the Americas (Appendix A [115,116,117,118,119,120,121,122,123,124,125,126,127,128]). Where possible, the current status of practice and implementation was confirmed with local stakeholders (Appendix A). There continues to be a disconnect between policy and practice (Figure 2). More than two thirds of the available national malaria treatment guidelines across the Asia-Pacific (13/17) include a statement on the need for G6PD testing, while testing is only implemented in four of those countries (Lao PDR, Thailand, and Myanmar using a PoC diagnostic (STANDARD G6PD, SD Biosensor, ROK), and centralized testing in South Korea). Despite the absence of a clear recommendation to test, PoC testing using the STANDARD G6PD is rolled out in Bangladesh, Cambodia, Vietnam, and the Solomon Islands, and is planned for 2023 in Bhutan. In addition, large feasibility studies including the use of the STANDARD G6PD are planned in Indonesia and Papua New Guinea.

In the Americas, only 4 of the 14 vivax endemic countries with available national antimalarial guidelines explicitly recommend G6PD testing (Brazil [116], Colombia [117], French Guiana [120] and Nicaragua [124]). Of these four countries with G6PD testing recommendations, only Brazil is implementing testing with the STANDARD G6PD in the context of the rollout of TQ. They are doing so with temporary approval of the device in two municipalities while it is under review by the National Committee for Health Technology Incorporation (Conitec). No information was ascertained for current implementation in French Guiana and Nicaragua, while Peru and Colombia are planning or conducting feasibility studies to inform the potential rollout. 

Out of the five available guidelines in Africa and the Middle East, only two (Somalia [112] and Madagascar [114]) recommend testing, although neither have implemented this in routine practice yet.

Treatment recommendations based on G6PD status vary between countries, and in most cases, there is a lack of guidance on how to treat patients with intermediate deficiency (Figure 3) or what to do if testing is unavailable.

Only a few of the guidelines reviewed included details on whether the deployed assay should be a PoC or laboratory-based assay, and whether the respective diagnostic should return a qualitative or quantitative result. Guidance on the latter was only indicated in Afghanistan [80,81], Lao PDR [91,92,93], Pakistan [98,99], Thailand [106], Vanuatu [107,108], Brazil [116], French Guiana [120], and Nicaragua [124]. Most guidelines do not provide the G6PD activity thresholds to define G6PD normal or deficient status. Some guidelines (11/17 from the Asia-Pacific and 8/14 from the Americas) use terminology such as “G6PD normal”, “intermediate”, and “deficient” without further clarification, while others (4/19 in the Asia-Pacific and 2/14 in the Americas) refer to the older nomenclature of “mild”, “moderate”, and “severe” status, again without a precise definition [129]. Only guidelines from Afghanistan [80,81], Lao PDR [91,92,93], Thailand [106], Vietnam [109,110], Brazil [116], Colombia [117], and Nicaragua [124] state what enzyme activity thresholds determine normal G6PD status either in percent activity or as an absolute value. Vietnam’s guidelines [109,110] provide thresholds but allow for qualitative or quantitative testing. None of the guidelines state what assay those thresholds are based on or how 100% G6PD activity is defined. 

## 5. Early Experiences with the Implementation of PoC G6PD Testing 

Several countries have begun implementing PoC diagnostics into routine care or plan to do so in the near future; most are using or considering the STANDARD G6PD. To date, the vast majority of STANDARD G6PD analysers and consumables (Figure 2) are distributed within the Asia-Pacific (58%) region, followed by the Americas (15%), Africa (14%), and the Middle East (11%), compared to only 2% in Europe (SD Biosensor, personal communication). Feedback from early experience with the use and implementation of the STANDARD G6PD in endemic countries (including Brazil, Bangladesh, Cambodia, Lao PDR, Thailand, and Vietnam) can be broadly divided into four types of considerations for the wider rollout: (i) the need for technical improvements of the device itself, (ii) logistical considerations, (iii) the training and supervision required, and (iv) the level of the healthcare system where the test can be performed [130,131,132,133,134,135,136].

Based on their experience with the STANDARD G6PD in Bangladesh, Cambodia, Lao PDR, and Vietnam, users identified required technical improvements and logistical considerations relevant to the wider rollout [130,131,132,133,134,137].

### 5.1. Technical Challenges 

In some areas, healthcare staff have prior experience with qualitative G6PD rapid diagnostic tests (RDTs), specifically the previously WHO-prequalified, qualitative CareStart G6PD lateral flow assay (Access Bio/CareStart, Somerset, NJ, USA) and the quantitative STANDARD G6PD (Table 1). These users appreciated the STANDARD G6PD’s easier result interpretation based on a numeric output and additional haemoglobin reading [130,131]. However, end users also indicated that the STANDARD G6PD testing procedure (Figure 4 [132]) was more complicated than the qualitative CareStart G6PD RDT, which is based on colorimetric principles (Table 1) [130,131]. Some end users reported difficulties with the required two pipetting steps, while others described difficulties with mixing the buffer with the blood sample [130,131,132]. In comparing procedural errors between regular test use and control runs, some users found running controls more difficult [137].

### 5.2. Logistical Considerations

End users commented that the number of included pipettes (n = 50 for 25 test devices) was insufficient to allow for procedural errors [130,133]. In malaria elimination settings with low case numbers, the original shelf life of the test devices (12 months) and original package sizes (25 strips per box) would result in a significant number of expired strips, wasting resources [130]. The shelf life of the test devices and buffer was regarded as being too short to accommodate local logistics, including import processes and customs clearances [130,131]. In response to this feedback, the manufacturer has increased the shelf life of the test devices to 18 months and now offers package sizes of 10 test devices per box (SD Biosensor, personal communication). Additionally, end users suggested making the machine rechargeable rather than being reliant on single-use batteries [130,131,132]. Feedback from community level users suggested that a percentage battery indicator on the screen would help identify devices with a low battery [132]. Community level feedback also recommended including a tube rack for the test kit buffer vials so that they do not fall over when working at village level without laboratory infrastructure [132]. Finally, the appropriate place and time for running controls is an important consideration. As a result of difficulties performing the quality control steps, in Lao, PDR controls have been phased out of district and health centre levels and are only conducted by the national program lab team during supervision visits [137]. In contrast, Cambodia maintains its practice of conducting control runs at the health centre level [130]. 

### 5.3. Training and Supervision

Adequate training and supervision are likely to be essential to the successful rollout of the STANDARD G6PD [77,138], especially in areas with low case numbers [134]. Standardised training materials [139] that include background information on the G6PD enzyme, G6PD deficiency, and test and control procedures as well as a practical training agenda have been developed [140]. Generally, NMCPs and pilot projects, such as those in Brazil, Vietnam, Lao PDR, and Bangladesh, have adapted these standardized training materials to varying degrees based on their country context [130,133,135]. National trainings have largely involved the trainings-of-trainers model, while in Lao PDR a training-of-trainers and cascade model starting with central level laboratory technicians was employed due to time and budgetary constraints [141]. 

Some key considerations for training are the ratio of trainee-to-trainer, time, and resource allocation, and the availability of qualified trainers [130]. An ideal trainer-to-trainee ratio has not been identified to date and varies by program. For the national rollout training in Cambodia, the ratio was 1 to 10, while in Lao PDR it was 1 to 8 at the provincial and district level [130]. In comparison, in research or pilot program contexts, Vietnam’s ratio was one to three and Brazil’s and Thailand’s were one to five [130,135,142]. Ensuring that sufficient time is allocated to individual practice is an important part of training [130,133,142]. Most training workshops of the STANDARD G6PD were delivered over half a day to a day, and in some cases were embedded into larger NMCPs case management training. Early experiences from Vietnam suggest that training at participants’ work sites would be more beneficial compared to centralised training, while in Cambodia, training laboratory technicians on the use of the G6PD test to act as facilitators during training was found to make the process easier [130]. During training, there were benefits to having a demonstration with visual aids, including using G6PD testing process video and an A3 size job aid [130,133,135]. 

Although users’ practical testing proficiency was the focus of most trainings, conducting formal practical assessments was dependent on time and capacity [133,135,136]. In some cases, formal practical assessments were conducted during supervision visits [136]. In Cambodia, the NMCP highlighted the need to have enough facilitators and time for the competency assessments during training—noting that not all trainees could be assessed [130]. These assessments are based on standardized materials tailored to a country’s needs to varying degrees [139]. 

Training of health care workers likely requires not only focusing on how to use the diagnostic but also how to interpret G6PD and haemoglobin readings and translate them into treatment decisions. The ongoing TQ Roll-out Study (TRuST) in Brazil and a pilot study in Lao PDR are currently assessing the provision of appropriate treatment according to a patient’s G6PD status, gender, and age [137,143]. 

In addition to training, adequate supervision of end users is critical, especially during the initial phases of the G6PD testing rollout [78,130,144,145]. Regular supervision visits are crucial for assessing and strengthening the capacity of health workers to conduct the test [130,134,145]. In Cambodia, the national program plans to conduct three supervision visits per year for 15 health centres with low scores in the post-training assessment [136]. In Lao PDR, an assessment is conducted during supervision visits, after which refresher training is provided if required [136]. The number of supervision visits are limited to regular case management supervision visits due to budgetary constraints. However, there is no clear guidance on what level of supervision is required, and this is likely dependent on overall health system capacity. In addition to regular supervision visits, refresher trainings are likely required with varying frequency, especially in areas with low malaria caseloads where trained health professionals use the test infrequently. In an operational study in Brazil implementing the STANDARD G6PD, the study location with the lowest case load had the lowest assessment score at 6 months after initial training. As such, the authors suggest refresher training every 6 months in such locations [146]. 

### 5.4. Considerations about Level of Health Care System 

Currently, most testing with the STANDARD G6PD is occurring at the health facility level; however, in most endemic settings, patients with malaria are diagnosed at the community level [147,148]. Hence current strategies for G6PD testing require referral of patients diagnosed with malaria at the community level to the health facility for G6PD testing and radical cure treatment. Successful routine referral from community health workers to health facilities with G6PD testing has proven to be challenging and limits the access to adequate treatment [130,136]. The proportion of vivax patients referred to health facilities for G6PD testing and radical cure treatment who subsequently present for testing at the referral site has been shown to be low [141,145]. In Cambodia between February and December 2021, 34% of patients referred by village malaria workers (VMWs) to health facilities for G6PD testing did not reach the health facility [149]. Similarly, in Lao PDR in 2021, only around a third of patients eligible for G6PD testing actually underwent testing, a discrepancy mostly attributed to challenges with patient referral [141]. Referral rates are likely to improve through enhanced training, outreach, patient education, and referral support [150]. This was demonstrated in Cambodia, where the use of qualitative and quantitative PoC G6PD testing was piloted before the wider rollout. In this study, all patients who were referred by VMWs completed referral and received G6PD testing at a health centre. Training of VWMs, community sensitisation, and VMWs often accompanying patients to the health centre aided in completing referrals [150]. 

An alternative strategy to referral is to conduct G6PD testing at community level. A pilot study from Cambodia found that VMWs have the capacity to perform G6PD testing if given appropriate training and supervision [132]. In the same study, VMWs received a one-day training course, including theoretical and practical components on how to use the STANDARD G6PD. Throughout the study, they were supported by a study coordinator and received refresher training during their monthly visit to the health centre. Comparison of VMW measurements with measurements conducted by laboratory technicians indicated no significant difference in absolute readings and a very good and significant correlation. Such findings are in line with Gerth-Guyette et al. (2021) who found that there was no significant difference in proficiency based on the level of laboratory experience [80]. However, there were differences in how G6PD results were interpreted between VMWs and laboratory technicians [151]. Although this study does show promise for VMWs conducting G6PD testing, quality of measurement and translation of STANDARD G6PD readings into treatment decisions warrants further investigation to inform the operationalisation of VMWs conducting G6PD testing. 

Based on these combined early experiences, several key themes need to be addressed as the rollout is progressing (Table 3).

## 6. Other Considerations for the Update of Novel Diagnostics

The early experiences with the STANDARD G6PD are encouraging, but reports are primarily focused on technical and logistical considerations. The slow uptake of similar PoC or RDTs such as those for malaria, HIV, and tuberculosis (TB) provide important lessons on the broader societal considerations that are often neglected, including economic implications for widescale implementation of the STANDARD G6PD or other PoC G6PD diagnostics [152].

Uptake of novel diagnostics is influenced by the regard for modern medicine in the community [152]. Apprehensions of new test formats likely contributes to users’ (patients and health workers) poor adherence to diagnostic results, as illustrated by the non-adherence to negative malaria RDT results and the over-prescription of anti-malarials seen in the early days of RDT introduction [152]. Additionally, social acceptance of a new technology is influenced by the perception of its risks and benefits by end users [153]. For example, the belief that blood is sacred influenced patients’ aversion to testing in the context of malaria diagnosis, a phenomenon likely to impact acceptance of any G6PD PoC as well [152]. Information and education communication therefore needs to be locally and culturally appropriate to address negative community perceptions of new diagnostic tests before they present a barrier, as well as provide relevant information on the benefits of the new technology (Table 3).

A shortage of qualified health workers and poorly developed medical infrastructure can contribute to test misuse, misdiagnosis, and test failure [152,154], ultimately leading to the erosion of user confidence in a new diagnostic device [152]. Supply chain issues are also an important infrastructural consideration that NMCPs have raised repeatedly [155] as a potential limiting factor for the widespread adoption of novel PoC diagnostics [152,154,156,157]. Reasons for interruptions of supply lines for diagnostics include inaccurate estimation of new shipment deliveries, insufficient preparation for seasonal demand, difficult transport conditions due to damaged roads, and inadequate compliance with inventory management practices at the local level [152]. In the context of introducing G6PD testing, it is therefore important to also consider supply chain and stock management as part of capacity building efforts, rather than singularly focusing on the end user performance of the test.

Historically, the absence of universal regulation and a standard evaluation process has slowed the uptake of new diagnostic tools such as malaria RDTs [152]. The WHO prequalification process is designed to provide adequate guidance for countries and donors to purchase quality assured products. As such, the Notice of Concern issued by the WHO Prequalification Team for the CareStart RDT has resulted in an interim halt of its rollout [158]. While the STANDARD G6PD is not yet WHO pre-qualified [159], it has received interim approval from the Expert Review Panel for Diagnostics (ERPD) [160], a mechanism aimed to review diagnostics that may have a high public health impact. The device has also received the Conformité Européene (CE) Mark (2017) and approval from the Australian Therapeutic Goods Administration (TGA; 2021) (SD Biosensor, personal communication). The ERDP approval has facilitated the purchase and rollout of the STANDARD G6PD in many countries (Appendix A) through the Global Fund. 

Finally, though PoC tests tend to be more affordable than alternative, more complex diagnostics [161], the cost per person screened with the STANDARD G6PD is relatively high (Table 1) [162,163]. These costs are unlikely to be covered by consumers, but rather by the health system and, by extension, external donors such as the Global Fund. Of particular concern is the cost of the STANDARD G6PD analyser per person screened, as this will be dependent on the device’s expected lifetime before it needs to be replaced and the number of patients screened per year. This means that the cost per person screened for G6PD deficiency will be higher at facilities that see fewer patients. In the context of the end stages of malaria elimination, it is unlikely that G6PD screening will be cost-effective in the short-term. Accordingly, the focus needs to shift to the longer-term benefits of malaria elimination. It is also important to note that the indicative price will not be the final cost, which may decrease after negotiation with the supplier after country-level regulatory approval but will need to include shipping, taxes, and other distribution expenses. Another related consideration when evaluating the economic impact of G6PD testing is deciding at what level(s) of the health system STANDARD G6PD might be placed. For example, placing STANDARD G6PD at hospitals that see more vivax malaria patients will result in lower costs per test administered than placing them at community facilities where fewer patients are seen. One option is to refer patients to higher level facilities, but this will be a challenge to balance with ensuring uptake given the findings from Cambodia that a third of patients did not use referrals to higher facilities [149].

The cost effectiveness of implementing G6PD screening with the STANDARD G6PD will vary significantly with heterogeneity in the underlying case burden and severity and the prevalence of G6PD deficiency. For example, in a country with mild variants and low prevalence of G6PD deficiency, the costs of screening may outweigh the benefits since haemolysis would be rare and unlikely to be severe. In places where primaquine is currently prescribed without G6PD screening, a value of information analysis could be conducted in order to help decide whether it is a better use of funding to conduct surveys to determine the prevalence of G6PD deficiency or whether G6PD screening should be implemented without this information.

Current malaria policies usually reflect associated funding by the Global Fund, so it is unclear whether the cost-effectiveness of G6PD testing is an important driver in country-level decision making. However national malaria control programs have ranked cost effectiveness of new approaches as a key area for further research [164], suggesting that this has some impact on policy decisions. Whether an intervention such as G6PD testing is considered cost-effective will also depend on the budget available, which could be a national health budget or a budget specifically for malaria control [165].

## 7. Conclusions

A range of G6PD PoC are now available, and some of these are likely to be WHO prequalified in the near future. Current experiences with the ongoing implementation of the STANDARD G6PD should inform the optimisation of the broader rollout of this test and will be crucial to facilitate the introduction of alternative diagnostic options. In many cases, policy guidance will need to incorporate the reality of implementation. 

## Figures and Tables

**Figure 1 pathogens-12-00650-f001:**
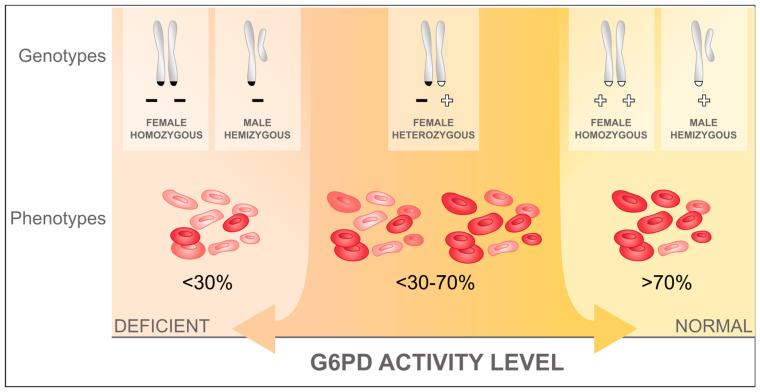
RBC populations in individuals with different G6PD alleles. RBCs with normal G6PD activity are coloured in red, while RBCs with deficient G6PD activity are pale red; adapted from Domingo et al. (2019) [33].

**Figure 2 pathogens-12-00650-f002:**
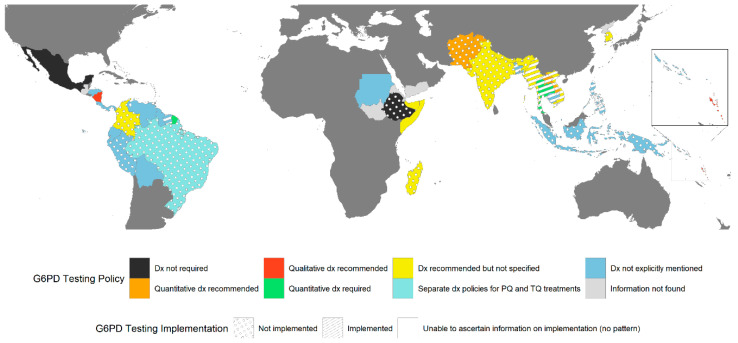
G6PD testing policies and their implementation in vivax endemic countries in the Asia-Pacific, the Horn of Africa and Madagascar, and the Americas. Different policies are indicated by different colours in the map, while the policy implementation status is indicated by the overlaying patterns.

**Figure 3 pathogens-12-00650-f003:**
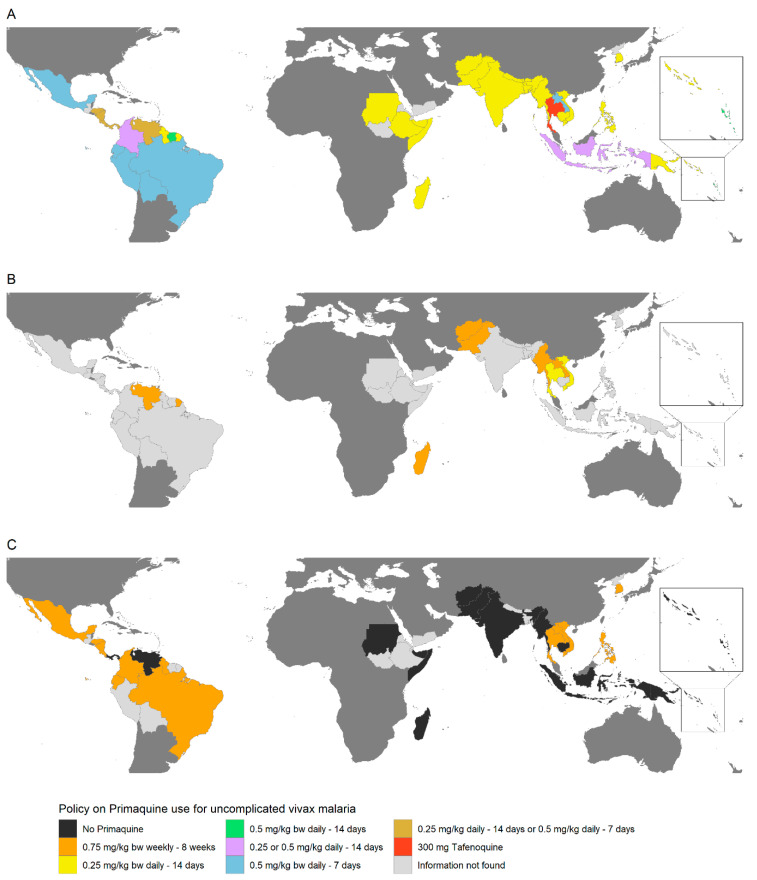
Policies on radical cure treatment for uncomplicated vivax malaria according to G6PD testing results in vivax endemic countries in the Asia-Pacific, the Horn of Africa and Madagascar, and the Americas. Treatment regimens are indicated by the country colour on the map. PQ treatment policies for G6PD normal, G6PD intermediate or mildly deficient, and G6PD deficient or severely deficient are shown in maps (**A**–**C**), respectively. bw = body weight. In Cambodia, G6PD normal individuals may also be given 0.75 mg/kg bw PQ weekly for 8 weeks. In Vanuatu and Panama, G6PD deficient individuals may also be given 0.75 mg/kg bw PQ weekly for 8 weeks, if access to supervision or blood transfusion is available.

**Figure 4 pathogens-12-00650-f004:**
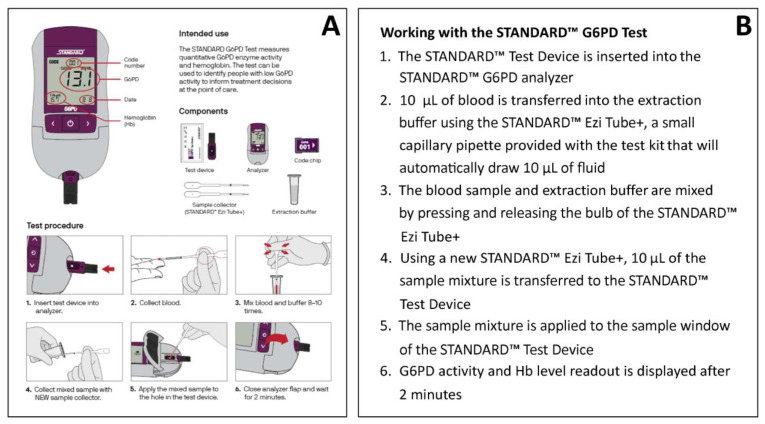
STANDARD G6PD Test Procedure: (**A**) Illustrated instructions for the use of the STANDARD G6PD Test from Adhikari et al. (2022) [132], and (**B**) Step-by-step detailed description of procedures.

**Table 1 pathogens-12-00650-t001:** Common phenotypic laboratory and PoC assays to detect G6PD deficiency.

Diagnostic to Detect G6PD Deficiency (Year First Reported)	Output	Blood Volume Required	Time to Result	Pipetting Steps in Sample Preparation	Cost *	Performance **
**Laboratory Assays**						
Spectrophotometry (1967) [48]	Quantitative	10 µL	15 min + calculation time	4–5 + sample or buffer preparation steps	Trinity Biotech (Ireland) USD 3.6 Pointe Scientific (USA) USD 2.0 [49]	Used as diagnostic reference; substantial inter-lab variability [34]
Fluorescent Spot Test (1966) [50,51]	Qualitative	10 µL	15 min + drying time	5	USD 0.1–3.0 [52]	At 30% AMM cut-off [52,53,54,55,56]: o Sens: 0.89–1.00 o Spec: 0.71–0.98
WST-8/1-methoxy PMS (2003) [57,58]	Quantitative or qualitative	5 µL	15–60 min	4	USD 0.1–3.2 [55,58]	Qualitative reading at 30% AMM cut-off [55]: o Sens: 0.84 o Spec: 0.98 Quantitative reading at 30% AMM cut-off [58]: o Sens: 0.55 o Spec: 0.98
Flow Cytometry (1989) [40,41,59]	Cytochemical	1 mL [59]	3 h	At least 14 + buffer preparation steps	USD 20 [60]	Discriminating heterozygous females ^$^: o Sens: 0.93 o Spec: 1.00 Discriminating homozygous females ^$^: o Sens: 1.00 o Spec: 0.98 Discriminating hemizygous males ^$^: o Sens: 1.00 o Spec: 0.97 [60]
**Point-of-care Assays**						
CareStart G6PD RDT (2011; AccessBio, Somerset, NJ, USA) [61]	Qualitative (2.7 U/g Hb threshold)	2 µL	10 min	2	USD 1.5 [62]	Pooled performance at 30% AMM cut-off [63]: o Sens: 0.96 o Spec: 0.95
BinaxNOW G6PD Test (2010; Alere, Waltham, MA, USA) [64,65]	Qualitative (4.0 U/g Hb threshold)	10 µL	7 min	3	USD 15 [66]	At 30% AMM cut-off [67]: o Sens: 1.00 o Spec: 0.995 At 60% median cut-off [64]: o Sens: 0.55 o Spec: 1.00 At 4 U/g Hb cut-off [65]: o Sens: 0.98 o Spec: 0.97
CareStart G6PD Biosensor (2017; AccessBio, Somerset, NJ, USA)	Quantitative	5 µL	4 min	0	USD 670 (device) + USD 3.4 (test strip) [68] USD 500 (device) + 2.5 (test strip) [58]	At 30% AMM cut-off [58,68,69,70,71]: o Sens: 0.06–1.00 o Spec: 0.99–1.00 At 70% AMM cut-off [68,70,71]: o Sens: 0.71–1.00 o Spec: 0.93–0.98
STANDARD G6PD Test (2018; SD Biosensor, Suwon, Republic of Korea)	Quantitative	10 µL	2 min	2	USD 380 (device) + USD 3 (test device) [68]	At 30% AMM cut-off [68,72,73]: o Sens: 1.00 o Spec: 0.97–0.99 At 70% AMM cut-off [68]: o Sens: 0.89 o Spec: 0.93 Females between 30% and 70% AMM cut-off [72,73]: o Sens: 0.82–0.97 o Spec: 0.88–0.97

Sens. = sensitivity; Spec. = specificity; AMM = adjusted male median; RDT = rapid diagnostic test. * Cost refers to manufacturer price recommendations and may vary significantly for different countries. ** Based on non-systematic literature review of commercial assays for which performance has been evaluated in the field, considering spectrophotometry as reference method. ^$^ Performance considering direct sequencing as reference method.

**Table 2 pathogens-12-00650-t002:** Key performance/agreement, storage requirement, and pricing characteristics as guided by the WHO’s TPP for future development of G6PD diagnostics [74].

Characteristics	Acceptable	Desirable
**TPP #1: PoC Screening Test for G6PD**		
Performance in percent positive agreement (PPA) or percent negative agreement (PNA)	Distinguish deficiency at 30% of normal G6PD U/g Hb threshold PPA ≥ 95% PNA ≥ 90%	Fulfil the acceptable criteria Able to distinguish intermediate G6PD activity at between ≥30% and <70% of normal G6PD U/g Hb threshold PPA ≥ 85% PNA ≥ 90%
Kit storage	18 months storage at 4–35 °C Humidity 75% + 5% Tolerates brief periods of >40 °C	24 months storage at 4–40 °C Humidity 85% + 5% Tolerates freezing and brief periods of >45 °C
Pricing	Test: <US $5.00 Instrument: <US $700.00	Test: < US $2.50 Instrument: <US $400.00
**TPP #2: One-time Quantitative Test for G6PD**		
Agreement	Systematic difference (bias): Absolute difference: ±2.0 IU/g Hb Fold difference: 0.8–1.2 fold Limits of agreement: Absolute difference: ±2.0 IU/g Hb Fold difference: 0.8–1.2 fold	Systematic difference (bias): Absolute difference: ±1.0 IU/g Hb Fold difference: 0.9–1.1 fold Limits of agreement: Absolute difference: ±1.0 IU/g Hb Fold difference: 0.9–1.1 fold
Kit storage	≥18 months storage at 4 °C Humidity 75% + 5% Tolerates freezing and brief periods >40–45 °C	≥24 months at 4 °C or ≥18 months at 18–25 °C Humidity 80% + 5% Tolerates freezing and brief periods >45 °C
Pricing	Test: <US $10.00 Instrument: <US $2000.00	Test: <US $5.00 Instrument: <US $1000.00

**Table 3 pathogens-12-00650-t003:** Key themes for G6PD test roll out that require further research.

**Themes to be** **addressed**	Best practice for training format Frequency of retraining Quality control and assurance of results in routine care Suitable level of health system End user proficiency and translation of readings into treatment decisions Optimal supply chain for test devices and controls Improvements of the test package (e.g., additional Ezi-tubes per kit) Culturally appropriate information and education communication for health workers and community to increase uptake

## Data Availability

Not applicable.

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
