# Peer review of "A Review of the Current Status of G6PD Deficiency Testing to Guide Radical Cure Treatment for Vivax Malaria"

_pathogens, 2023, doi:10.3390/pathogens12050650_

Round 1

Reviewer 1 Report

This review covers comprehensive information related to G6PD diagnostic, effectiveness and policies. Content-wise is good. Suggest to include some information/findings from patent search related to G6PD diagnostics.

Minor grammatical errors or unclear sentence structure
Line 141 remove 'once'

Line 159-160 There continues......

Wordings in Figure 2 is blur, need to increase the clarity

Line 176-179 'Of these only Brazil...' consider revise this sentence. it is long, unclear and confusing

Wordings in Figure 3 is blur, improve resolution

Line 272 & 297 "rollout" should be roll out. Please standardize the term through out the article

Table 3, right column - justified left side

Author Response

Response to comments from Reviewer 1

Point 1. This review covers comprehensive information related to G6PD diagnostic, effectiveness and policies. Content-wise is good. Suggest to include some information/findings from patent search related to G6PD diagnostics.

Response 1. This review focuses on diagnostics that are commercially available. Considering future assays would go beyond the scope of this review.

Minor grammatical errors or unclear sentence structure:

Point 2. Line 141 remove 'once'

Response 2. Line 153: The “once” is needed to indicate that one-time testing was proposed, from which the result would guide treatment decisions going forward.

Point 3. Line 159-160 There continues......

Response 3. We have checked the manuscript thoroughly and corrected for any typing and grammatical errors.

Point 4. Wordings in Figure 2 is blur, need to increase the clarity

Response 4. We have replaced Figure 2 in the manuscript with a high-resolution version.

Point 5. Line 176-179 'Of these only Brazil...' consider revise this sentence. it is long, unclear and confusing

Response 5. We have changed the respective paragraph for clarity. Line 188-192: Of these four countries with G6PD testing recommendations only Brazil is implementing testing with the STANDARD G6PD in the context of TQ roll out. They are doing so with temporary approval of the device in two municipalities while it is under review by the National Committee for Health Technology Incorporation (Conitec).

Point 6. Wordings in Figure 3 is blur, improve resolution

Response 6. 

We have replaced Figure 3 in the manuscript with a high-resolution version.

Point 7. Line 272 & 297 "rollout" should be roll out. Please standardize the term through out the article

Response 7. We have changed “rollout” to “roll out” throughout the manuscript.

Point 8. Table 3, right column - justified left side

Response 8. All bullet points in Table 3 are now left justified.

Reviewer 2 Report

Is there any information about the TQ secondary effects and how long it lasts? I suggest including it if so.

Line 71. I suggest to include how the G6PD is important to maintain the cell's redox potential in terms of enzymatic reaction and product(s) of this for a better understanding of the G6PD function at molecular level.

Technical challenges. You mention some common technical issues with the STANDARD device as double pipetting, what about the process of validation of the analyzer similar to the Quality Control process  Line 253? 

Regarding the cost and effectiveness In countries where PQ was the treatment por Pv hyponozoites, is it worth measuring G6PD deficiency knowing the AHA is very rare? Is there any information about that? Please add if so.

Author Response

Response to comments from Reviewer 2

Point 1. Is there any information about the TQ secondary effects and how long it lasts? I suggest including it if so.

Response 1. The G6PD deficiency-related side effects of TQ are discussed in lines 67-76.

Point 2. Line 71. I suggest to include how the G6PD is important to maintain the cell's redox potential in terms of enzymatic reaction and product(s) of this for a better understanding of the G6PD function at molecular level.

Response 2. We have added a sentence providing more details about the molecular mechanisms of the G6PD enzyme. Line 79-84: In red blood cells (RBCs), G6PD is essential to maintain the cells’ redox potential by producing reduced NADPH [24]; reduced NADPH is a key electron donor for the conversion of oxidised glutathione (GSSG) into reduced glutathione (GSH. Reduced GSH in turn captures free radicals that could cause oxidative damage. Human RBCs contain neither a nucleus nor mitochondria and cannot replenish G6PD levels.

Point 3. Technical challenges. You mention some common technical issues with the STANDARD device as double pipetting, what about the process of validation of the analyzer similar to the Quality Control process  Line 253? 

Response 3. We have updated the respective sentences to highlight concerns around quality control in both the technical challenges and logistical consideration sections. Line 251-253: In comparing procedural errors between regular test use and control runs, some users have found running controls more difficult [137]. Line 273-277: Finally, the appropriate place and time for running controls is an important consideration. As a result of difficulties performing the quality control steps, in Lao PDR controls have been phased out of district and health centre levels and are only conducted by the national program lab team during supervision visits [137].

Point 4. Regarding the cost and effectiveness In countries where PQ was the treatment por Pv hyponozoites, is it worth measuring G6PD deficiency knowing the AHA is very rare? Is there any information about that? Please add if so.

Response 4. There is no study yet that measures the value of G6PD testing in this context. We have added a sentence starting in line 439 that highlights how value of information analyses could help to address this: In places where primaquine is currently prescribed without G6PD screening, a value of information analysis could be conducted to help decide whether it is a better use of funding to conduct surveys to determine the prevalence of G6PD deficiency or whether G6PD screening should be implemented.

Reviewer 3 Report

Overall Comments :

The paper deals with a highly relevant topic pertinent to malaria diagnosis and treatment and is important from the malaria e, limitation targets also. The treatment of P.vivax malaria for radical cure using  8-aminoquinolines causes hemolysis in glucose-6-phosphate dehydrogenase (G6PD) deficient patients and in absence of an effective WHO prequalified diagnostic tool approved for G6PD deficiency testing, the cure of vivax malaria has become challenging. As per WHO, ‘At present, some patients with P. vivax malaria receive only chloroquine, leaving them vulnerable to repeated relapses. Also, in many regions, a high proportion of P. vivax malaria prevalence can be attributed to relapses following the activation of dormant hypnozoites, which contribute to onward transmission. Expanded access to P. vivax radical cure is critical to reducing the malaria burden and reaching malaria elimination targets.

Though the overall paper is written well, the authors need to review the paper thoroughly for finalization and address gaps in specific sections in the paper in addition to other minor revisions. The interpretation of the cited references is found to be incorrect at times and needs special attention and care to ensure that the intended meaning is conveyed appropriately and the inferences drawn in these papers are not misquoted or wrongly expressed( e.g., Introduction ). Several figures and maps have been reproduced in the paper from existing published sources, but the source has not been mentioned or acknowledged. It is recommended that all such sources should be appropriately mentioned under each figure in parentheses and appropriate citations should also be provided in the text. The number of references can also be reduced appropriately e.g., instead of three references for the burden of P.vivax malaria in the first sentence of the introduction the WMR 2021 reference would suffice.   Avoid citing the reference number instead of the author and year, wherever applicable. Some facts and figures need to be rechecked e.g., the radical cure for Pv as recommended by WHO. Please consider adding a map to show the global G-6-PD distribution since the testing policies in different regions have been shown. The conclusion needs to be a bit more informative than it is now to inform the readers better in a concise manner about the status and limitations of the WHO-recommended G-6-PD point of care tests, their limitations, merits and demerits. In short can this paper be a guide to the readers regarding the decision on selection of the point of care G-6-PD test selection backed by WHO recommendations and country policies?

Specific comments about different sections are provided below for further review and revision of the paper:

Specific Comments:

Introduction:

Line 36-38: The paper is being published in 2023. Please provide the latest P.vivax burden data here as per the World Malaria Report of 2021. Though a reference to the World Malaria Report, 2021 has been provided, the text reads ‘the global P. vivax burden reduced from 24.5 million in 2010 to 14.3 million in 2017’.

Line 39: This line starts with ‘Despite these advances’. This is sort of ambiguous. The preceding text in the paper doesn’t mention any advances.

Line 40-44: Please revise the text for better clarity and to ensure that the intended meaning is conveyed. The authors write  ‘P. vivax gametocytes develop before the onset of symptoms, resulting in asymptomatic but infectious patients who cannot be identified by passive surveillance [4-6]. Secondly, the sensitivity of most point-of-care (PoC) tests for P. vivax is suboptimal since P. vivax causes peripheral bloodstream infections at lower parasite densities than P. falciparum [7-9]’.

Contrary to the above statements, all the papers cited deal with asymptomatic infections wherein the authors have tried to highlight the issue of asymptomatic and submicroscopic P. vivax infections being gametocyte carriers thereby indicating that asymptomatic infections might significantly contribute to malaria transmission in this region. Issues of diagnosis could be pertinent to low parasitemia as well as the asymptomatic nature of cases as these carriers are not picked up by active as well as passive surveillance due to lack of symptoms.  The issue of a sensitive diagnostic tool has also been highlighted in the cited studies. However, while talking about the issues related to the diagnosis and detection of asymptomatic malaria and the diagnostic tools it is pertinent to write carefully because asymptomatic infections are found in both Pf and Pv and both are likely to contribute to a hidden reservoir of transmission.

Line 49-59: Please cite a WHO reference regarding the 8-AQ guidance i.e., administered over 14 days as a total dose of 3.5 mg/kg when used for radical cure. Also, mention whether it is an adult dose and what are the restrictions. The information appears incorrect as well as incomplete. WHO recommends the following  for radical cure of P.vivax malaria :

Table 1: This table is a bit jumbled up. The first column doesn’t have a heading. Please separate the performance characteristics and references into separate columns. Also, add the year to show when a particular test was made available. Add a footnote to describe all abbreviations, numbers etc. to make it more comprehensive and easier to understand. Ensure that the information is systematic and easy to understand.

Table 2. The performance/agreement, storage requirement, and pricing characteristics as guided by  the WHO’s TPP for future development of G6PD diagnostics

It is not clear why select items are covered under this. Some important facts and points are missing and the information is not complete as given in the referred WHO document(69). There are important considerations and situations also covered by WHO which are lacking here. The table needs to be reviewed and revised to ensure that the correct message is conveyed and important points related to these TPPs are not missed, being interrelated e.g., intended use, type of setting, the limit of quantification etc.

Moreover, just listing is not enough. Discussion is required on important parameters.

Policy and guidelines :

Supplementary tables have been referred to. Are these going to be part of the published paper with available to the readers? If not, please review and revise to include the important information in the main text of this paper. Please add references for lines 174-196 to show the source of information.

References: Old references and guidelines of WHO etc. conveying numbers and figures are seen in the paper. These should be replaced in the main body as well as under the references heading by the latest references and figures. I am not pointing out all the references one by one here and am leaving it to the authors to revise this part meticulously.

Conclusion: Please summarize the important achievements and limitations of the G-6-PD point of care testing in addition to what has been given for benefit of the readers and researchers.

Author Response

Response to comments from Reviewer 3

Point 1. Checking that all figures from other sources are appropriately referenced in figure titles and main text.

Response 1. We have reviewed the entire text. We added references to Figures 1 and 4, while Figures 2 and 3 were created by us based on the information cited in Supplementary Tables 1-3. We have also added a reference in lines 95 and 248 (References [33] and [132]).

Point 2. Reducing the number of references for some of the main points in the introduction.

Response 2. Following the logic of a review, we aim to cite source documents rather than other reviews. Accordingly, the number of cited articles is quite high.

Point 3. Avoid citing the reference number instead of the author and year, wherever applicable.

Response 3. We are following the in-text citation format as specified by Pathogens and have added in author and year when publications are directly referred to in the text e.g. “Figure 1. RBC populations in individuals with different G6PD alleles. RBCs with normal G6PD activity is coloured in red, while RBCs with deficient G6PD activity is a pale red colour; adapted from Domingo et al. (2019) [33].”

Point 4. Please consider adding a map to show the global G-6-PD distribution since the testing policies in different regions have been shown.

Response 4. We cite the key article that presents the global prevalence of G6PD deficiency (Howes et al, 2012, 2013, and 2013 [27-29]. The focus of this review is around the key policies and practices of G6PD point of care testing globally and not so much on the global prevalence of G6PD deficiency. We have therefore not included the maps created by Howes et al.

Point 5. Line 36: instead of three references for the burden of P.vivax malaria in the first sentence of the introduction the WMR 2021 reference would suffice.

Response 5. Line 37-40: There is variability in the estimate of global P. vivax prevalence over time. We have therefore included the key articles on the topic as well as the WHO World Malaria Report 2022.

Point 6. Line 36-38: The paper is being published in 2023. Please provide the latest P.vivax burden data here as per the World Malaria Report of 2021. Though a reference to the World Malaria Report, 2021 has been provided, the text reads ‘the global P. vivax burden reduced from 24.5 million in 2010 to 14.3 million in 2017’.

Response 6. We have revised this section to include both numbers from World Malaria Report 2022, which presents the most updated estimations, and from Battle et al. 2019 [2], whose estimations are more all-encompassing than WHO-reported cases.

Line 37-40: According to the World Health Organisation (WHO) cases decreased from 24.5 million in 2000 to 4.9 million in 2021[3]; while Battle et al. estimated a reduction of 41.6% between 2000 and 2017, from 24.5 to 14.3 million cases [2].

Point 7. Line 39: This line starts with ‘Despite these advances’. This is sort of ambiguous. The preceding text in the paper doesn’t mention any advances.

Response 7. Line 42: “The advances” refers to the reduction in vivax burden in the previous sentences (line 37-40).

Point 8. Line 40-44: Please revise the text for better clarity and to ensure that the intended meaning is conveyed. The authors write  ‘P. vivax gametocytes develop before the onset of symptoms, resulting in asymptomatic but infectious patients who cannot be identified by passive surveillance [4-6]. Secondly, the sensitivity of most point-of-care (PoC) tests for P. vivax is suboptimal since P. vivax causes peripheral bloodstream infections at lower parasite densities than P. falciparum [7-9]’.

Contrary to the above statements, all the papers cited deal with asymptomatic infections wherein the authors have tried to highlight the issue of asymptomatic and submicroscopic P. vivax infections being gametocyte carriers thereby indicating that asymptomatic infections might significantly contribute to malaria transmission in this region. Issues of diagnosis could be pertinent to low parasitemia as well as the asymptomatic nature of cases as these carriers are not picked up by active as well as passive surveillance due to lack of symptoms.  The issue of a sensitive diagnostic tool has also been highlighted in the cited studies. However, while talking about the issues related to the diagnosis and detection of asymptomatic malaria and the diagnostic tools it is pertinent to write carefully because asymptomatic infections are found in both Pf and Pv and both are likely to contribute to a hidden reservoir of transmission

Response 8. We have changed the wording in the second sentence for more clarity and to emphasize the difference between vivax and falciparum. Line 43-49: P. vivax gametocytes develop before the onset of symptoms, resulting in asymptomatic but infectious patients who cannot be identified by passive surveillance [4-6]. Secondly, the sensitivity of most point of care (PoC) tests for P. vivax is lower than P. falciparum due to lower parasite densities [7-9].

Point 9. Line 49-59: Please cite a WHO reference regarding the 8-AQ guidance i.e., administered over 14 days as a total dose of 3.5 mg/kg when used for radical cure. Also, mention whether it is an adult dose and what are the restrictions. The information appears incorrect as well as incomplete. WHO recommends the following for radical cure of P.vivax malaria :

Response 9. We have revised the sentence to clarify that this is in line with WHO recommendations (with citation [15]) and added another sentence to reflect the latest addition in the most updated version of the guideline [16]. The contraindications relating to G6PD deficiency is elaborated in the next paragraph, and country-specific recommended doses are discussed in Section 4. Line 58-61: Primaquine (PQ) is the most widely used 8-AQ [14] and is usually administered over 14 days as a total dose of 3.5 mg/kg when used for radical cure in line with WHO recommendations [15]. More recently, the WHO Treatment Guidelines were updated to include a 7-day treatment regimen with the same total dosage [16].

Point 10. Table 1: This table is a bit jumbled up. The first column doesn’t have a heading. Please separate the performance characteristics and references into separate columns. Also, add the year to show when a particular test was made available. Add a footnote to describe all abbreviations, numbers etc. to make it more comprehensive and easier to understand. Ensure that the information is systematic and easy to understand.

Response 10. We have added a heading for the first column, and the year when the diagnostic was first reported. The explanation of abbreviations at the bottom of the table has been updated. We did not separate the performance information and respective references into different columns because some performance indicators have different sources, and the current format offers more clarity.

Point 11. Table 2. The performance/agreement, storage requirement, and pricing characteristics as guided by  the WHO’s TPP for future development of G6PD diagnostics. It is not clear why select items are covered under this. Some important facts and points are missing and the information is not complete as given in the referred WHO document(69). There are important considerations and situations also covered by WHO which are lacking here. The table needs to be reviewed and revised to ensure that the correct message is conveyed and important points related to these TPPs are not missed, being interrelated e.g., intended use, type of setting, the limit of quantification etc. Moreover, just listing is not enough. Discussion is required on important parameters.

Response 11. We present key acceptable and desirable characteristics that are relevant to the scope of our review paper and cite the source document to allow the reader to review the full WHO document. To clarify we have amended the title of Table 2, which now reads: “Key performance/agreement, storage requirement, and pricing characteristics as guided by the WHO’s TPP for future development of G6PD diagnostics”. The context of Table 2 is discussed in lines 150-155.

Point 12. Supplementary tables have been referred to. Are these going to be part of the published paper with available to the readers? If not, please review and revise to include the important information in the main text of this paper. Please add references for lines 174-196   to show the source of information.

Response 12. As is standard practice, all supplementary files that are included in the submission will be published along with the main article.

Point 13. Old references and guidelines of WHO etc. conveying numbers and figures are seen in the paper. These should be replaced in the main body as well as under the references heading by the latest references and figures. I am not pointing out all the references one by one here and am leaving it to the authors to revise this part meticulously.

Response 13. We have updated the WHO reports/guidelines citations /references to the latest versions [3, 15, 16]. If there is any reference to earlier versions, it is content-based and not referring to numbers.

Point 14. Please summarize the important achievements and limitations of the G-6-PD point of care testing in addition to what has been given for benefit of the readers and researchers. The conclusion needs to be a bit more informative than it is now to inform the readers better in a concise manner about the status and limitations of the WHO-recommended G-6-PD point of care tests, their limitations, merits and demerits. In short can this paper be a guide to the readers regarding the decision on selection of the point of care G-6-PD test selection backed by WHO recommendations and country policies?

Response 14. This descriptive review paper summarizes the latest knowledge base of G6PD deficiency testing, policy, and implementation and is summarized by the abstract. The objective of this article is not to offer recommendations on test and treat algorithms. We have also presented key themes to be addressed in rolling out G6PD testing (Table 3) based on all the sources we reviewed in this paper.